# Prevalence and Factors Associated with Sarcopenia in Patients on Maintenance Dialysis in Australia—A Single Centre, Cross-Sectional Study

**DOI:** 10.3390/nu13093284

**Published:** 2021-09-20

**Authors:** Marille Umakanthan, John Wing Li, Kamal Sud, Gustavo Duque, Daniel Guilfoyle, Kenneth Cho, Chris Brown, Derek Boersma, Muralikrishna Gangadharan Komala

**Affiliations:** 1Department of Renal Medicine, Nepean Hospital, Kingswood, NSW 2747, Australia; marille.umakanthan@health.nsw.gov.au (M.U.); john.li@health.nsw.gov.au (J.W.L.); kamal.sud@health.nsw.gov.au (K.S.); daniel.guylfoyle@health.nsw.gov.au (D.G.); kenneth.cho@health.nsw.gov.au (K.C.); derek.boersma@health.nsw.gov.au (D.B.); 2Nepean Clinical School, The University of Sydney, Sydney, NSW 2006, Australia; gustavo.duque@unimelb.edu.au (G.D.); c.brown@sydney.edu.au (C.B.); 3Australian Institute for Musculoskeletal Science (AIMSS), The University of Melbourne and Western Health, St Albans, VIC 3021, Australia

**Keywords:** sarcopenia, dialysis, elderly, Australia

## Abstract

**Background:** Sarcopenia is associated with significant morbidity and mortality in patients with chronic kidney disease. The prevalence of sarcopenia in the dialysis population varies from 4% to 63%. However, the prevalence and risk factors of sarcopenia in the Australian dialysis population remain uncertain. **Aim:** To study the prevalence of sarcopenia in patients on maintenance dialysis by using the European Working Group on Sarcopenia in Older People (EWGSOP) diagnostic criteria of sarcopenia and to identify associated risk factors. **Methods:** We evaluated adult patients on maintenance haemodialysis and peritoneal dialysis in this single-centre cross-sectional study in Australia. Patient’s clinical (age, gender, dialysis modality and diabetic status) and laboratory parameters (serum albumin, calcium, phosphate, 25-hydroxy-vitamin D and parathyroid hormone levels) were investigated. We employed bioimpedance spectroscopy, hand grip dynamometer and the timed up and go test (TUG) to evaluate muscle mass, strength and function, respectively. **Results:** We evaluated 39 dialysis patients with a median age of 69 years old. The prevalence of sarcopenia was 18%. Sarcopenia was associated with low serum albumin (*p* = 0.02) and low serum phosphate level (*p* = 0.04). Increasing age and female sex were potential risk factors for sarcopenia (*p* = 0.05 and 0.08, respectively). Low lean muscle mass, reduced hand grip strength and prolonged TUG were present in 23.1%, 41% and 40.5%, respectively, of the cohort. The hand grip test had good correlation with lean muscle evaluation and the TUG. **Conclusions:** Sarcopenia was prevalent in 18% of maintenance haemodialysis patients from an Australian single-centre cohort, with low serum albumin and phosphate as significant risk factors.

## 1. Introduction

Sarcopenia has historically been considered as an age-related phenomenon and plays a major role in frailty, disability and mortality in the geriatric population [1,2]. In fact, sarcopenia is now recognised as a progressive and generalised skeletal muscle condition involving loss of skeletal muscle mass and function in the latest International Statistical Classification of Disease and Related Health Problems—Australian Modification [3,4]. Studies of the elderly population associate primary sarcopenia with several factors such as vitamin D deficiency and hyperparathyroidism [5,6]. Secondary causes of sarcopenia, particularly those that are disease related, are increasingly being recognised. Chronic kidney disease (CKD), being a hypercatabolic state, is regarded as an important driver of muscle wasting, resulting in significant morbidity and mortality in this population, thus the association between the two entities has been vigorously investigated in the recent era.

Our understanding of the pathogenesis and factors associated with sarcopenia in CKD patients is evolving into an overarching concept of uraemic sarcopenia [5]. In addition to the uraemic milieu, the drivers of muscle atrophy (primarily type IIB fibre) and dysfunction in such a state include (1) abnormal muscle mitochondrial function from renin–angiotensin–aldosterone system malfunction; (2) deficiencies in testosterone, oestrogen and growth hormone; (3) mechanical changes from physical inactivity, protein energy wasting and comorbid conditions; (4) activation of the ubiquitin–proteasome pathway and (5) changes in muscle protein balance [5]. The recent literature also suggested sarcopenia is closely associated with raised inflammatory markers, which is common in CKD patients [6,7].

The prevalence of sarcopenia in patients with end-stage kidney disease and its associations with clinical markers has been partially explored [6,7]. In 2016, 12,706 patients received dialysis in Australia, with 9263 patients (72.9%) aged more than 55 years old [8]. The prevalence of sarcopenia in the Australian dialysis population is unknown.

In this first cross-sectional analytical study in Australia, we aim to study the prevalence of sarcopenia in patients on maintenance dialysis by using the European Working Group on Sarcopenia in Older People (EWGSOP) diagnostic criteria of sarcopenia and to identify associated risk factors. We also aim to test the correlation between practical assessment tools for muscle strength and function against muscle mass.

## 2. Materials and Methods

### 2.1. Study Design and Participants

We recruited all adult patients aged 18 or above, on maintenance haemodialysis and peritoneal dialysis, at a satellite dialysis centre of a tertiary referral university teaching hospital in Western Sydney, Australia, from 1st May 2015 to 30th June 2016. Exclusion criteria included amputees and cognitive impairment which limited study participation. 

We recorded patients’ clinical parameters (age, sex, dialysis modality and diabetic status) and laboratory parameters (serum albumin, calcium, phosphate, 25-hydroxy-vitamin D (25OH-D) and parathyroid hormone (PTH) levels) at the date of study entry. All subjects underwent assessment of muscle mass, muscle strength and muscle function. Patients were defined as sarcopenic if they had low muscle mass in addition to either low muscle strength or low muscle function, as per EWGSOP [9]. The study protocol was approved by the local ethics committee (Study 15/31-HREC/15/NEPEAN/69).

### 2.2. Muscle Mass

We measured muscle mass indirectly by Bioimpedance Spectroscopy (BIS) (Body Composition Monitor (BCM), Fresenius, Germany) and presented it as lean tissue index (LTI). We defined an LTI of less than the 10th percentile for an age- and sex-matched population as a marker of reduced muscle mass [10]. LTI measurements were performed pre-dialysis in haemodialysis patients and at random for peritoneal dialysis patients. Details of BCM measurement were outlined by the manufacturer online [11].

### 2.3. Muscle Strength

We assessed muscle strength by Jamar Handgrip dynamometer with patients’ handgrip strength recorded in a standardised posture of upright seating, arms by the side with the elbow at a 90-degree angle. Three readings were taken from both hands and the best reading was recorded. We defined reduced muscle strength as handgrip strength measurement below 26 kg in men and 16 kg in women [12].

### 2.4. Muscle Function

We assessed patient’s muscle function by the timed up and go test (TUG). We defined reduced muscle function in patients with TUG scores above the 95% confidence interval of their age-matched normal population scores. Abnormal TUG tests were defined as greater than 9 s, greater than 10.2 s and greater than 12.7 s for patients 60–69 years, 70–79 years and 80–99 years of age, respectively [13].

### 2.5. Statistical Analysis

Baseline characteristics were summarized by count and percentage, means with standard deviation (SD) or medians with interquartile ranges (IQR) where appropriate. Logistic regression was used to test the association between sarcopenia and clinical covariates, and we report the odds ratio. No adjustments were made for multiple comparisons. Analyses were undertaken in SPSS v24 (IBM, Chicago IL USA). Correlations between the three muscle parameters were found using the Pearson correlation coefficient (GraphPad Prism Version 6). A two-sided *p*-value less than 0.05 was considered statistically significant. 

## 3. Results

### 3.1. Baseline Characteristics

We recruited 39 dialysis patients, and all completed the evaluation. Baseline clinical and laboratory characteristics of the patient population are summarized in Table 1. The median age of this population was 69 years (IQR 54–77), and 72% of the patients were men. In total, 62% of patients were on haemodialysis and 31% of the patients had diabetes. Patients on haemodialysis were dialysed for a minimum of 4.5 h, thrice weekly. Amongst patients on peritoneal dialysis, 13 were on automated peritoneal dialysis and 2 patients were on continuous ambulatory peritoneal dialysis.

Baseline laboratory parameters showed serum albumin 32.4 ± 3.7 g/L, calcium 2.4 ± 0.2 mmol/L and phosphate 1.5 ± 0.6 mmol/L. Median PTH and 25OH-D level were 31.8 pmol/L (IQR 22.8–85.4) and 66 nmol/L (IQR 43.8–78.3), respectively. 

### 3.2. Indicators of Sarcopenia

In total, 23% of the studied population had reduced LTI by BIS with a mean LTI of 14.1 ± 3.2 kg/m^2^. We detected reduced handgrip strength in 41% of the study group with a median grip strength of 27.0 ± 12.3 kg. In total, 41% of the study population had elevated age-matched TUG, thus regarded as a decrease in muscle function. The median TUG was 8 s (IQR 6–15 s).

### 3.3. Prevalence of Sarcopenia and Associated Factors

The prevalence of sarcopenia by EWGSOP definition was 18% (7 out of 39 patients). Associated factors are summarized in Table 2. Patients with sarcopenia tended to be older (77 years old versus 64, OR 1.16, *p* = 0.05) and of female sex (4 of 11 females versus 3 of 28 males were sarcopenic, OR 4.76, *p* = 0.08). There was no significant association with dialysis modality and presence of diabetes. For laboratory parameters, sarcopenic patients had lower serum albumin (29.3 g/L versus 33.1 g/L, OR 0.72, *p* = 0.02) and lower serum phosphate level (1.08 mmol/L versus 1.6 mmol/L, OR 0.08, *p* = 0.04). They had slightly higher calcium, PTH and 25OH-D levels compared to their non-sarcopenic counterparts, but the association between these parameters and sarcopenia did not reach statistical significance. There was no association between inflammatory parameters, CRP or dialysis vintage on sarcopenia.

### 3.4. Correlation between Muscle Parameters

Correlations between the three muscle function tests were evaluated in pairs, that is, (a) LTI and handgrip strength, (b) LTI and TUG and (c) handgrip and TUG. Handgrip strength correlated well with both LTI and TUG (*p* < 0.05). However, no correlation was noted between LTI and TUG (*p* = 0.17).

## 4. Discussion

In this first cross-sectional study on patients on maintenance dialysis from Australia, we found that the prevalence of sarcopenia was 18% by the EWGSOP definition. The prevalence of sarcopenia in the population aged greater than 60 years has been noted to be approximately 10% in both men and women [14]. Hence, the prevalence of sarcopenia in our dialysis population is clearly higher than that of the age-matched general population. However, the prevalence is not as high as reported from previous studies in patients on dialysis of age 50 or above [6,12]. 

The prevalence of sarcopenia is quite variable across studies looking at sarcopenia in similar populations. Socioeconomic status is one important determining factor, as Dorosty et al. illustrated a marked difference of 12.8% compared to 20.8% between higher- and lower-income elderly participants [15]. The variability is also due to a non-uniform definition of sarcopenia, leading to a wide interpretation of its prevalence [16,17]; non-uniform reference cut-offs for the domains of sarcopenia such as muscle mass and muscle performance; and a lack of a standardized method for muscle mass assessment [18]. We used the EWGSOP definition of sarcopenia in our study and believe this universally accepted definition would form a solid foundation for standardisation in future studies of sarcopenia in the geriatric population, and a better understanding of the prevalence in the dialysis population [9]. In the “Kidney Function and Sarcopenia in the United States General Population NHANES III” (2007), using bioimpedence to estimate muscle mass and identify sarcopenic patients showed a high prevalence of 60% in patients with glomerular filtration less than 60 mL/min [19]. This estimate is likely to be different to the revised definition of sarcopenia, as it now incorporates low muscle function or strength in addition to low muscle mass. Muscle mass alone is insufficient to determine sarcopenia and the relationship between strength and mass is non-linear [9]. This is supported by studies of sarcopenia in the haemodialysis population, where Marcus et al. showed a weak correlation between measurements of muscle atrophy and function [20], and Isoyama et al. demonstrated that low muscle strength was more strongly associated with ageing, protein-energy wasting, physical inactivity, inflammation and mortality than low muscle mass in the haemodialysis population [21]. The EWG guidelines recommend gait velocity to determine muscle function. In our study, we adopted TUG as a more simplified and appropriate measure due to frailty and comorbidities within our population.

We identified that low serum albumin and phosphate levels were significantly associated with sarcopenia in our dialysis population, while there was also a trend showing age being associated with sarcopenia. Our study is consistent with other studies in the dialysis and healthy older population where age is an important factor in determining sarcopenia. Nutritional status and serum albumin have been found to correlate with muscle function in the dialysis population [21]. Low albumin and phosphate are reflectors of poor oral intake and protein energy wasting [22] and have been shown to be independent risk factors for mortality on dialysis as well [23,24]. 

We also identified a higher incidence of sarcopenia amongst female patients, although this did not reach statistical significance. There is scarce literature on the differential prevalence of sarcopenia between genders. Physical inactivity has been noted to be higher in females compared to males and there is a correlation between physical inactivity and sarcopenia in patients with chronic kidney disease [25]. Therefore, by extrapolation, we hypothesize that it is the physical inactivity of female dialysis patients that led to sarcopenia, rather than other clinical factors. The direct relationship remains to be elucidated. 

Surprisingly, we did not find a correlation between PTH or 25OH-D levels with sarcopenia. This contrasts with previous studies, which have noted an association between vitamin D and PTH levels with sarcopenia in the general population. Vitamin D deficiency is associated with sarcopenia in the elderly population [5,26] and vitamin D supplementation in deficient sarcopenic haemodialysis patients has been shown to increase muscle size and strength [27]. Secondary hyperparathyroidism was also found to have an associative role [28,29]. The lack of association in our study could be due to a different definition of sarcopenia, as we adopted the newer EWGSOP guidelines. Moreover, most of the subjects had appropriate medications to achieve optimal 25OH-D and PTH levels, hence eliminating differential levels as a factor and probably contributing to the lower incidence of sarcopenia in our study population. We also did not find a correlation between dialysis modality and sarcopenia. This was confirmed in a previous study [30].

We identified a good correlation between handgrip and LTI and TUG. Handgrip strength has been found to be a reproducible tool for identifying sarcopenia, as more muscle mass tends to be reflected as stronger muscle power [18]. However, the correlation between TUG and LTI is poor. A standard TUG is a simple and practical test which takes less than ten seconds and involves walking 3 m. It is more associated with a patient’s functional mobility and static and dynamic balance, resulting in the assessment of risk of fall. In fact, TUG is a good screening tool for sarcopenia [31]. TUG and gait velocity test both predict most geriatric outcomes [32]. However, the correlation between TUG and muscle mass is unclear and demands further studies to verify. We therefore suggest, in addition to bioimpedance measurement, handgrip strength assessment to be an appropriate, simple and cheap method to perform in routine clinical practice to screen for sarcopenia in the dialysis population.

Our study has a number of strengths. To our knowledge, this is the first cross-sectional analysis of sarcopenia amongst the dialysis population in Australia using the EWGSOP definition of sarcopenia. Our study comprises both maintenance haemodialysis and peritoneal dialysis patients. The prevalence of sarcopenia was lower in our population, and this may be due to a lower prevalence of vitamin D deficiency and significant hyperparathyroidism; this could also be a limitation, as this prevents us from identifying these as possible factors of sarcopenia in this population. We can identify patients on dialysis at high risk of sarcopenia by medical record review and simple assessment tools, including BIS and a handgrip dynamometer. This will enable population-targeted preventive measures, namely regular dietary review and exercise physiotherapy advice to maintain muscle strength and function. Early intervention targeting risk factors may reduce the incidence of sarcopenia and improve the quality of life in these patients. 

Nevertheless, there are a few limitations of note. We used an indirect measurement of lean muscle mass by using bioimpedance spectrometry. Although it is accepted as a practical method to reflect muscle mass, there is a paucity of data to demonstrate good correlation with the gold standard measurement of skeletal muscle mass by dual X-ray absorptiometry. Moreover, this was a single-centre study. The study population was relatively small in number, and a large proportion of them were on peritoneal dialysis at home. They are potentially more well-nourished as they are independent in the home environment and more motivated when participating in various tests, which is in contrast with the wider population in the region, where there is a higher proportion of diabetes requiring in-centre monitoring and medical care while undergoing dialysis. Even though dialysis modality did not influence outcomes in this study, the small study population may not have been enough to identify a difference in outcomes. Nonetheless, we demonstrated that even in such a population, 18% were sarcopenic, which warrants further evaluation by future multicentre trials with larger sample sizes. We did not collect data on the use of phosphate binders or records of dialysis adequacy. However, the serum phosphate and PTH levels were reasonable, which might be suggestive of acceptable dialysis adequacy. 

## 5. Conclusions

In conclusion, the prevalence of sarcopenia was 18% in this single-centre cross-sectional study by using the EWGSOP definition for patients on maintenance dialysis in Australia. In addition, we report low serum albumin and phosphate levels were associated with sarcopenia. Individuals of advancing age and females tend to have a higher risk of sarcopenia. We demonstrated that handgrip strength measured by dynamometer as an indicator of muscle strength was a good marker for sarcopenia and correlated well with muscle mass and muscle function, and should be an integral part of the assessment of sarcopenia in this population. The TUG test did not correlate with lean tissue index in our study. Future research should focus on larger studies in the dialysis population, and on evaluating whether interventions such as intensive dietetic education and exercise physiotherapy prescription could reduce the incidence of sarcopenia, a common but silent disease of modern society. 

### 5.1. Practical Application

Sarcopenia affected 18% of our dialysis population, where low serum albumin and phosphate levels were associated risk factors. Bioimpedance spectroscopy and handgrip dynamometer are feasible methods for the earlier identification of sarcopenia and to guide potential interventions.

### 5.2. Compliance with Ethical Standards

All procedures performed in this study were in accordance with ethical standards of the Nepean Blue Mountains Local Health District (NBMLHD) ethics committee (Study 15/31-HREC/15/NEPEAN/69) and with the 1964 Helsinki Declaration and its later amendments or comparable ethical standards. Informed consent was obtained from all participants. There was no funding for the project and there was no conflict of interest for any of the authors.

## Figures and Tables

**Table 1 nutrients-13-03284-t001:** Baseline characteristics of all patients on maintenance dialysis.

Characteristics	Overall Cohort
*n* (%) Median age, year ^1^	39 (100) 69 (54, 77)
Male (%)	28 (72)
**Dialysis modality**	
Haemodialysis (%)	24 (62)
Peritoneal dialysis (%)	15 (38)
Diabetes mellitus	12 (31)
Serum albumin ^2^ (g/L)	32.4 ± 3.7
Serum calcium ^2^ (mmol/L)	2.4 ± 0.2
Serum phosphate ^2^ (mmol/L)	1.5 ± 0.6
Serum parathyroid hormone ^1^ (pmol/L)	31.8 (22.8, 85.4)
Serum 25 hydroxy-vitamin D ^1^ (nmol/L)	66 (43.8, 78.3)

Legend: ^1^—interquartile range; ^2^—standard deviation.

**Table 2 nutrients-13-03284-t002:** Risk factors for sarcopenia in Australian patients on maintenance dialysis.

Characteristics	Sarcopenic (*n* = 7)	Non Sarcopenic (*n* = 32)	Odds Ratio	*p* Value
Median age (IQR)	77 (69, 80)	64 (19,84)	1.16	0.05
Female sex (%)	4 (36)	3(11)	4.76	0.08
Dialysis modality—HD (%)	5 (71)	19 (59)	1.71	0.56
Diabetes (%)	2 (29)	10 (31)	1.14	0.89
Serum albumin (g/L) ^1^	29.3 ± 1.8	33.1 ± 3.6	0.72	0.02
Serum calcium (mmol/L) ^1^	2.41 ± 0.09	2.37 ± 0.16	4.80	0.55
Serum phosphate (mmol/L) ^1^	1.08 ± 0.32	1.59 ± 0.55	0.08	0.04
Serum PTH (pmol/L) ^2^	42.5 (4.8,315)	30 (0.0–235)	1.00	0.46
Serum 25 OH-D (nmol/L) ^2^	68 (22–87)	65 (24–179)	0.99	0.68
CRP	8 (1–19)	7 (1–62)	1.01	0.91
Months on dialysis	53 (2–108)	34 (3–144)	1.01	0.25

Legend: HD—haemodialysis; PTH—parathyroid hormone; 25OH-D—25 hydroxyvitamin D; CRP—C-reactive protein; ^1^—standard deviation; ^2^—range lowest to highest.

## Data Availability

The data presented in this study are available as a Appendix A.

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
