# Peer review of "Prevalence and Factors Associated with Sarcopenia in Patients on Maintenance Dialysis in Australia—A Single Centre, Cross-Sectional Study"

_nutrients, 2021, doi:10.3390/nu13093284_

Round 1

Reviewer 1 Report

Dear Dr. Gangadharan Komala,

Thank you very much for the opportunity in reviewing the manuscript.

Regarding a cross-sectional study examining the prevalence and associated risk factors of sarcopenia in maintenance dialysis patients (hemodialysis (HD) and peritoneal dialysis (PD), authors found that the prevalence of sarcopenia was 18% and hypoalbuminemia as well as hypophosphatemia were associated with higher risk of sarcopenia. Also, muscle strength was correlated with muscle mass and function.

The following are comments and suggestions.

- Since this study is a single-center retrospective study with a small number of study population which were less likely to represent maintenance dialysis population in Australia, conclusion in the abstract with 18% prevalence of scarcopenia in Australian maintenance dialysis may be too strong. Suggest changing wording to “Prevalence of sarcopenia is 18% among maintenance dialysis patients from an Australian single-center cohort” or other wording that is not indicate over representation of Australian maintenance dialysis population.

- Laboratories were tested at the time of study entry. Were these blood sample collected before, during, or after HD or PD.

- HD and PD patients may have different characteristics related to sarcopenia and clinical outcomes. It would be interesting to stratify all study population by dialysis modality and report and/or analyze the outcomes (sarcopenia) and risk factors of sarcopenia if the number of patients in each category allows. If not, please discuss in limitation.

- In a paragraph discussing about strength of this study, authors mentioned that a low prevalence of sarcopenia in this study was due to lower prevalence of vitamin D deficiency and significant hyperparathyroidism. This should be discussed as one of the limitations.

Author Response

Dear Reviewer

Many thanks for your valuable suggestions which add value to our manuscript. We have attached below the responses to your queries.

Kind Regards

Dr Muralikrishna Gangadharan Komala

Reviewer 2 Report

Dear Sir

About the manuscript entitled “Prevalence and factors associated with Sarcopenia in Patients on Maintenance Dialysis in Australia- A single centre, cross sectional study”, by Umakanthan M et al:

The authors used the “European Working Group on Sarcopenia in Older People (EWGSOP)” diagnostic criteria to evaluate the presence of sarcopenia (EWGSOP’). This is a correct methodology.

But:

  • About the population included:

The number of patients enrolled is low, and the total number of patients dialyzed at the moment this cross-sectional study was performed is not provided.

 An exclusión criterion was cognitive dysfunction. But mild cognitive dysfunction is highly prevalent in dialysis patients, not only because of renal failure but also because most patients are older. 

The authors should clarify how many patients over the total dialyzed population in the center were included. They have also to specify what was considered “cognitive impairment which limited study participation” as exclusión criteria  (¿mild, moderate, grave?)

  • About the clinical parameters included

No inflammation parameter was included, though the authors mentioned in the discussion that inflammation is highly prevalent in dialysis patients, and it favors the presence of sarcopenia.  Reactive C protein is frequently registered in dialysis centers to evaluate inflammation, and it correlates with the presence of sarcopenia.  I suggest RCP or another inflammation parameter be included in the analysis.

  • No data about the time the patients had been under dialysis treatment were provided. This is important also, as time under dialysis is an independent risk factor for sarcopenia

Even though the number of patients evaluated for sarcopenia is low, provided the corrections suggested are fulfilled.

Author Response

Dear Reviewer

Many thanks for your valuable suggestions to our manuscript. We have made appropriate corrections and believe the manuscript has benefitted from your review. Please find attached below our answers to your recommendations.

Kind Regards

Dr Muralikrishna Gangadharan Komala

Reviewer 3 Report

The Article is devoted to the study of the prevalence and risk factors of sarcopenia in patients on maintenance dialysis in the single centre in Australia. The current study is highly important as sarcopenia may lead to physical disability, poor quality of life and increased mortality in older adults and at the same time chronic kidney disease is considered to cause muscle wasting and subsequent morbidity and mortality in this population. European Working Group on Sarcopenia in Older People (EWGSOP) diagnostic criteria of sarcopenia were used.

It is also a strong part of the study design that assessment of muscle strength, function and mass was aimed, as it is agreed by the different specific consensuses on sarcopenia that diminished muscle mass in connection with decreased muscle function is obligatory to diagnose sarcopenia.

At the same time, probably, it would be noteworthy to include in the Article a bit detailed information about the patients’ diets, if it is possible. As it is known that diet and nutrition may prevent and manage sarcopenia.

Authors chose an interesting approach based on using of appropriate, simple and cheap methods which could be applied in routine clinical practice to screen for sarcopenia. Association of biochemical parameters levels (serum albumin, serum phosphate) with sarcopenia are demonstrated. Potential risk factors, such as advancing age and female sex, are pointed out.

The cohort, described in the study, contains 39 patients (median age 69-years-old) with sarcopenia on maintenance dialysis. Despite the fact, that the number of patients in the research is not very high, which Authors also pointed out, and there were patients with diabetes in the chosen population, which, probably, requires additional explanation, current study might represent a starting point, that could navigate further, more detailed research, which will help to characterise Australian dialysis population, because prevalence and risk factors of sarcopenia in this population remain unclear.

The following question does not diminish the value of the Article, it is just to specify the Article aim and design. Data, represented in the Article, were collected during the period from 2015 to 2016, probably, it might be interesting to add data which appeared after 2016? Would it be possible?

Data are statistically processed, which gives an opportunity to plan further research.

Probably, it is worthy to show the results of the muscle mass, muscle strength and muscle function assessment, so the definition of the patients as sarcopenic would be clear? Or probably it might be possible to include this information in supplementary materials?

The Article contains practical application which is a useful additional information.

The following comments do not diminish the importance of the Article:

Is it possible to include the information from section “2. Objective” in section “Introduction” to comply with the requirements posted on the Journal’s website? And, in this case, it would be necessary to correct sections numeration.

It would be also better to provide Author contributions to the work.

According to the Journal’s requirements, reference numbers should be placed in square brackets in the text.

It is also necessary to check the references description according to the requirements posted on the Journal’s website.

It would be better to rename the section Methods to the “Materials and methods” according to the Journal’s requirements.

Line 75 It is necessary to add a dot after the following part of the sentence: “muscle strength or low muscle function as per EWGSOP 9”.

Line 95 It is necessary to add a space between numbers and text in the following part of the sentence: “60-69years”.

Line 109 It would be better to start the following sentences with the text: “62% of patients were on haemodialysis”, “31% of”, if possible.

Line 113. It is necessary to add a space between numbers and text in the following part of the sentence: “3.7g/L”.

Lines 113, 114 It would be better to put the units equally through the text in the following parts of the sentences: “32.4 ± 3.7g/L”, “2.4 mmol/L ± 0.2mmol/L”, “1.5±0.6 mmol/L”.

Line 114 It is necessary to add a space between numbers and text in the following part of the sentence: “0.2mmol/L”

Line 119 It would be better to start the following sentence with the text: “23% of studied population”, if possible.

Line 121 It would be better to start the following sentence with the text: “41% of study population”, if possible.

Lines 127, 130, 131 Probably it would be better to describe the differences between the characteristics’ values (median age, serum albumin, serum phosphate levels) in the text avoiding re-mentioning the information which is presented in Table 2?

Line 128 Could you please specify, what does ratio “3/28”, presented in the text, mean? This indicator is also placed in the column “Non sarcopenic”, line Female sex (%), in Table 2. According to the Table 2, it should, probably, mean an amount of non sarcopenic female patients, but it is not clear how the indicator value was obtained.

Line 130 It is better to add spaces between numbers and text in the following parts of the sentences: “29.3g/L”, “33.1g/L”, “1.08mmol/L”.

Line 162 It is better to add a space between numbers and text in the following part of the sentence: “60ml/min”.

Line 193 It is important to check the font size of the following text “sarcopenic haemodialysis patients has been shown to increase muscle size”.

Line 238 It would be better to start the title of section 6 with a capital letter.

Author Response

Dear Reviewer

Many thanks for your valuable recommendations. We have accepted your recommendations and conducted the appropriate additional analysis. Our answers are enclosed below.

Kind Regards

Dr Muralikrishna Gangadharan Komala
